# Distinct and shared B cell responses of tuberculosis patients and their household contacts

**Komal Singh[1], Rajesh Kumar[1], Fareha Umam[1], Prerna Kapoor[2], Sudhir Sinha[1]\*, Amita Aggarwal[1]\***

**1** Department of Clinical Immunology and Rheumatology, Sanjay Gandhi Postgraduate Institute of Medical Sciences, Lucknow, India, **2** DOTS Centre, Sanjay Gandhi Postgraduate Institute of Medical Sciences, Lucknow, India

\* aa.amita@gmail.com (AA); sinha.sudhir@gmail.com (SS)

**Data Availability Statement:** All relevant data are within the paper and its Supporting Information files.

**Funding:** This study was funded by an ad-hoc research grant (No. 5/8/5/60/ITRC/adhoc/2018/

## Abstract

This study was aimed at identifying the B cell responses which could distinguish between 'latent tuberculosis infection (LTBI)' and active TB disease. Study subjects were smear-positive TB patients (n = 54) and their disease-free household contacts (HHCs, n = 120). The sera were used for determination of antibody levels (ΔOD values) against *Mycobacterium tuberculosis* membrane (MtM) antigens by ELISA and for visualisation of seroreactive MtM antigens by immunoblotting. B cell subsets in whole blood samples were determined by flow cytometry. In TB sera, levels of IgG antibodies were significantly higher than IgM and IgA whereas IgM and IgA antibody levels were comparable. Conversely, HHC sera had significantly higher IgM antibody levels than IgG and IgA. The ratio of IgM to IgG antibodies in HHCs were also significantly higher than in patients. Immunoblotting revealed that some of the MtM antigens (<10, ~12 and ~25 kDa) reacted with TB as well as HHC sera whereas some other antigens (~16, ~36, ~45 and ~60 kDa) reacted with most of TB and a subset of HHC sera. Frequencies of classical memory B cells (cMBCs, CD19+CD27+) were significantly higher, and of IgG+ cMBCs were significantly lower in HHCs than in patients. Frequencies of IgA+ cMBCs in HHCs and patients were comparable but both were significantly higher than the corresponding frequencies of IgG+ cMBCs. Frequencies of IgA+ atypical MBCs (aMBCs, CD19+CD27-) in HHCs and patients were also comparable and significantly higher than the IgG+ aMBCs. The plasmablast (CD19+CD27++CD38++) frequencies in HHCs and patients were comparable. These results suggest that the IgM/IgG antibody ratio, antibody binding to selected MtM antigens and relative frequencies of MBC subsets could indicate protective or pathogenic immune responses following the primary infection with Mtb. Responses that orchestrate protection leading to a 'quiescent' LTBI may provide clues to an effective vaccination strategy against TB.

ECD-1) of Indian Council of Medical Research (icmr.gov.in) to AA. The funders had no role in study design, data collection and analysis, decision to publish, or preparation of the manuscript.

# Introduction

Nearly 10 million people developed tuberculosis (TB) and 1.4 million died from TB in 2019 [1]. This situation, coupled with increasing incidences of treatment failure, has reemphasised the need for an effective vaccine. Although cell-mediated immunity (CMI) is believed to play a key protective role against TB, performances of BCG and other vaccines targeting the CMI have been below the expectations [1, 2] suggesting that other components of the immune system could also be important. An early immune response to *Mycobacterium tuberculosis* (Mtb) limits the infection to an asymptomatic state known as 'latent TB infection (LTBI)'. Effectiveness of this response is evident from the fact that < 10% of the latently infected persons develop TB in their lifetime [3]. An insight into the immune mechanisms underpinning LTBI may therefore provide clues to the components involved in protection against TB.

Accumulating evidences suggest that B cells may also play a significant role in protection against TB [4, 5]. B cells of the lung granuloma can process and present antigens to T cells, secrete antibodies and modulate inflammation [6]. Persons infected with Mtb show an abundance of circulating antibodies to Mtb antigens and the available evidences suggest that these antibodies, depending on their nature and specificity, could play an important role in pathogenesis of, or protection against TB [6]. The role of antibodies in protection against TB has been highlighted by us [7, 8] as well as others [9–11]. We have shown that opsonizing antibodies present in the sera of latently infected individuals can restrict the survival of Mtb within host macrophages by enhancing the process of phagosome maturation [8]. In an elegant study, passive immunization with a recombinant human IgA antibody to Mtb alpha-crystallin antigen, in combination with mouse IFN-γ, inhibited the Mtb infection in mice transgenic for human FcαRI [12]. However, it is not clear whether the isotypes of naturally produced antibodies can modulate the course of infection [4, 13].

Titres of circulating antibodies against Mtb reflect the quantum of the underlying infection [6]. Even so, there is a remarkable person-to-person variation in the antigens recognized by these antibodies [14]. In a major study [15], sera of TB suspects recognized approximately 10% of the Mtb proteome whereas antibody response in active TB cases was restricted to only about 0.5% of the proteome. In this and other studies including our own [7–9, 16], Mtb membrane (MtM) has emerged as a major contributor to the B cell antigenome. Immunoblotting represents a hypothesis-free approach to study the B cell antigenome and several workers have used it to visualise the Mtb antigens recognized by sera from TB patients and other study groups, including LTBI [13, 17–21]. However, the results have mostly been inconclusive due mainly to the differences in study populations or study protocols. A study from India had performed immunoblotting with the cell wall, membrane and cytosol of Mtb to show that cell wall and membrane contained most of the seroreactive antigens [19].

Generation of immunological memory is the hallmark of adaptive immunity. Classical memory B cells (cMBCs) express CD27 and derive their efficacy from the class-switched B cell receptors (BCRs) comprising IgG and IgA [22]. However, a subset of class-switched MBCs, designated as 'atypical' MBCs (aMBCs), does not express CD27. The aMBCs have fewer somatic hypermutations, show reduced proliferation and express antibodies of a broad specificity [23]. Upon antigen re-challenge, the MBCs differentiate into antibody secreting plasmablasts (PBs) defined as CD27$^{bright}$ CD38$^{bright}$ cells [22]. Not much is known about the B cell phenotypes which could distinguish between LTBI and active TB disease. Available reports are based on either flow cytometry which determines the frequency of circulating B cell subsets irrespective of their antigen specificity, or ELISPOT assay which counts antibody secreting cells (ASCs) specific for the selected antigen(s). However, ASC counts could vary depending on the selected antigens [24]. Another limitation of ELISPOT method is that the MBCs need

to be pre-activated *in vitro* in order to convert to ASCs. In the flow cytometry-based studies, frequencies of cMBC were found significantly higher and those of PBs significantly lower in LTBI than in active TB [9, 25, 26]. In addition, Zimmermann et al [9] showed that antibodies produced by PBs preferentially targeted the MtM antigens. In ELISPOT based studies also, the frequency of antigen-specific MBCs was higher and PBs was lower in LTBI than in active TB [24, 25, 27]. However, reports on class-switched cMBCs or aMBCs are scanty, even contradictory [25, 28].

The available literature suggests that B cell responses are differently modulated in active TB disease and LTBI. However, there is insufficient clarity on the measurable outcomes of these differences. To revisit these differences, we have conducted this study in a cohort of smear-positive TB patients and their household contacts (HHCs). HHCs of smear-positive patients have shown a greater hazard rate for the development of culture-positive TB than the HHCs of smear-negative patients or non-contacts [29]. Therefore, they are likely to provide definitive clues to the B cell responses associated with quiescent or progressive states of LTBI. As outcome measures we have determined the levels as well as antigen recognition patterns of anti-MtM antibodies present in the sera, and MBC and PB subsets of B cells present in the peripheral blood of study subjects.

## Methods

### Ethics statement

The study protocol was approved by Institutional Ethics Committee of Sanjay Gandhi Postgraduate Institute of Medical Sciences (SGPGIMS), Lucknow [IEC Code: 2017-197-EMP-99 (B)]. All participants provided a written informed consent and the study was conducted in accordance with the Declaration of Helsinki.

### Study subjects

The study subjects (n = 174) comprised a cohort of pulmonary TB patients (TB, n = 54) and their household contacts (HHCs, n = 120). All patients had active disease as defined by WHO [30], with their sputum-smears positive for acid-fast bacilli. All had drug-sensitive disease as confirmed by Xpert MTB/RIF assay [31] and all were seronegative for HIV. The HHCs were living with respective index cases (TB patients) for >1 year, fulfilling and exceeding the WHO criteria for the definition of a 'TB contact' [32]. Subjects with following conditions were excluded: Pregnant and lactating women, existence of secondary immunodeficiency conditions such as diabetes mellitus, organ transplantation, malignancy, treatment with corticosteroids and HHCs with any sign or symptom suggestive of TB.

### Tuberculin skin test

TST was performed in HHCs by following the recommendations of IUATLD [33] and WHO [34]. Accordingly, 5 tuberculin units (TU) of PPD (Arkray Healthcare, Surat, India) were injected intradermally on the left forearm. Mean diameter of the induration was measured after 48–72 hours and an induration of $\geq$ 10 mm was considered as positive.

### Blood samples

Blood was collected by standard venipuncture in sodium heparin tubes (2 ml) for B cell subset assays and in plain tubes (3 ml) to prepare sera for antibody assays.

## Mtb antigen

Mtb cell membrane (MtM) was isolated using a previously reported protocol [35]. In brief, 3 week old culture of Mtb (strain H37Ra, ATCC 25177) on Lowenstein-Jensen medium was harvested and bacterial sediment washed and re-suspended in PBS (0.2 g wet wt/ ml). The cell lysate obtained by sonication was centrifuged (23,000g) to settle unbroken cells and cell wall debris. Supernatant was re-centrifuged (150,000g) to obtain the membrane (sediment) and cytosol (supernatant). Membrane was washed and reconstituted with PBS and protein was estimated using a modified Lowry's method suitable for membrane proteins [16]. All protein aliquots were stored at -80 ˚C. As a quality control measure, identity of Mtb was validated periodically by immunochromatographic detection of MPT64 antigen [16].

## ELISA

A previously described protocol [16] was used with some modification. Antigen (10 µg/ml MtM protein) in coating-buffer (0·05 M carbonate, pH 9·5) or coating-buffer alone was dispensed (50 µl/well) in U-bottom ELISA plates (Nunc Maxisorp) and incubated overnight at 4˚C. After washing with tris-buffered saline (0·05 M Tris, 0·1 M NaCl, pH 7·4) containing 0·05% Tween 20 (TBS-T), the plates were incubated (90 min, 37˚C) with blocking solution (2% skimmed milk powder dissolved in TBS-T, 100 µl/well). After removing the blocking solution, test sera (diluted 1:500 in 1% milk-TBS-T) were dispensed in antigen- and buffer-coated wells (50 µl/ well, in duplicate) and incubated (90 min, 37˚C). Plates were later washed with TBS-T and incubated (50 µl/well, 90 min, 37˚C) with affinity-purified peroxidase-conjugated antibodies (Sigma) to human IgG, IgA or IgM (diluted 1:4000 in 1% milk-TBS-T). Plates were finally washed with TBS-T and the substrate solution (TMB, Bio Rad) was added (50 µl/well) and incubated at room temperature (20 min, in dark). Reaction was stopped by adding 7% $H_2SO_4$ (50 µl/well). Optical densities (ODs) were read at 450 nm in a plate reader (BioTek Synergy H1). For each serum, difference in the mean ODs of antigen and buffer coated wells was expressed as ΔOD.

## SDS-PAGE and immunoblotting

SDS-PAGE was performed on 12.5% resolving gel in a mini Protean-II cell (Bio Rad). The resolved antigens were electroblotted onto nitrocellulose paper (NCP, Millipore) in a mini trans-blot cell (Bio Rad). The antigen (MtM) and molecular weight markers (PAGE Ruler Prestained Protein Ladder, Thermo Scientific, Cat No. 26616) were run side-by-side in the same gel. After electroblotting, area corresponding to the antigen was cut into strips. Individual strips were probed with TB and HHC sera as follows. Strips were blocked (2 h, RT) with 2% BSA-TBS-T (2% bovine serum albumin in 50 mM Tris, 100 mM NaCl, 0.05% Tween 20; pH 7.4) and incubated (2 h, RT) with sera diluted (1:100) in 1% BSA-TBS-T. After extensive washing with TBS-T, the strips were incubated (2 h, RT) with alkaline phosphatase-conjugated antibody to human IgG (Sigma A3187), diluted (1:2000) in 1% BSA-TBS-T. After another round of washings, strips were incubated with the substrate solution (BCIP/NBT, Sigma B5655). Colour reaction was stopped by washing with water. Blot images were acquired on ImageQuant LAS 500 (GE Healthcare).

## Quantification of B cell subsets

The previously described protocols [22, 23] were followed. Briefly, blood samples were incubated with a combination of fluorescent labelled antibodies to human CD19 (PE-labelled, clone SJ25C1, BD), CD27 (APC-labelled, clone M-T271, BD), CD38 (PerCPCy5.5-labelled,

clone HIT2, BD), IgG (PECy7-labelled, clone G18-145, BD) and IgA (FITC-labelled, clone M24A, Sigma). RBCs were lysed with the lysis buffer (BD) and approximately $10^5$ WBCs were acquired on a flow-cytometer (FACS Canto-II, BD). From the lymphocytes gated on forward and side scatter, B cells were gated with CD19 staining. Data were analysed with FloJo software (Tree Star) to determine the frequencies of following B cell types: classical memory (CD19 +CD27+), switched classical memory (CD19+CD27+IgA+/IgG+), atypical memory (CD19+-CD27-IgA+/IgG+) and plasmablast (CD19+CD27++CD38++).

## Statistical analysis

The results are expressed, wherever indicated, as median with inter-quartile range (IQR). Differences between datasets were computed by either Wilcoxon matched pair signed rank test (for paired data) or Mann-Whitney test (for unpaired data). Correlations between datasets were computed by Spearman's rho. P values < 0.05 were considered as significant. All statistical analyses were performed by using GraphPad Prism 7 software.

## Results

### Demography of study subjects

Demographic characteristics of the index cases (TB patients) and their HHCs are summarised in Table 1. All patients had active, smear-positive disease (Bacterial Index 1+ to 3+) and none had drug-resistant TB or co-infection with HIV. All HHCs were close relatives (husband/ wife/ mother/ father/ son/ daughter) of the index cases, living in the same household for >1 year.

### Serum antibody levels against MtM antigens

Serum IgG, IgM and IgA antibody levels against MtM are shown in Fig 1A. Among the patients (n = 54), levels of IgG (median ΔOD, IQR: 1, 0.65–1.3) were significantly higher than

**Table 1. Baseline characteristics of study subjects.**

| Category | Characteristics | Output |
|---|---|---|
| Index cases | | |
| | Number recruited (female) | 54 (20) |
| | Median age (with IQR) in years | 27 (21–45) |
| | Pulmonary TB[1] | 54/54 |
| | Drug-resistant TB[2] | 0/54 |
| | HIV co-infection[3] | 0/54 |
| | Treatment prior to recruitment[4] | <3 Weeks |
| Household contacts[5] | | |
| | Number recruited (female) | 120 (68) |
| | Median age (with IQR) in years | 30 (22–42) |
| | Tuberculin test positive (%)[6] | 54/120 (45) |

[1]Diagnosed by chest x-ray and sputum-smear microscopy.

[2]Determined by GeneXpert.

[3]Determined by ELISA.

[4]Treatment as per WHO guidelines.

[5]Continuous living (for >1 year) in the household with index case.

[6]≥10 mm skin induration.

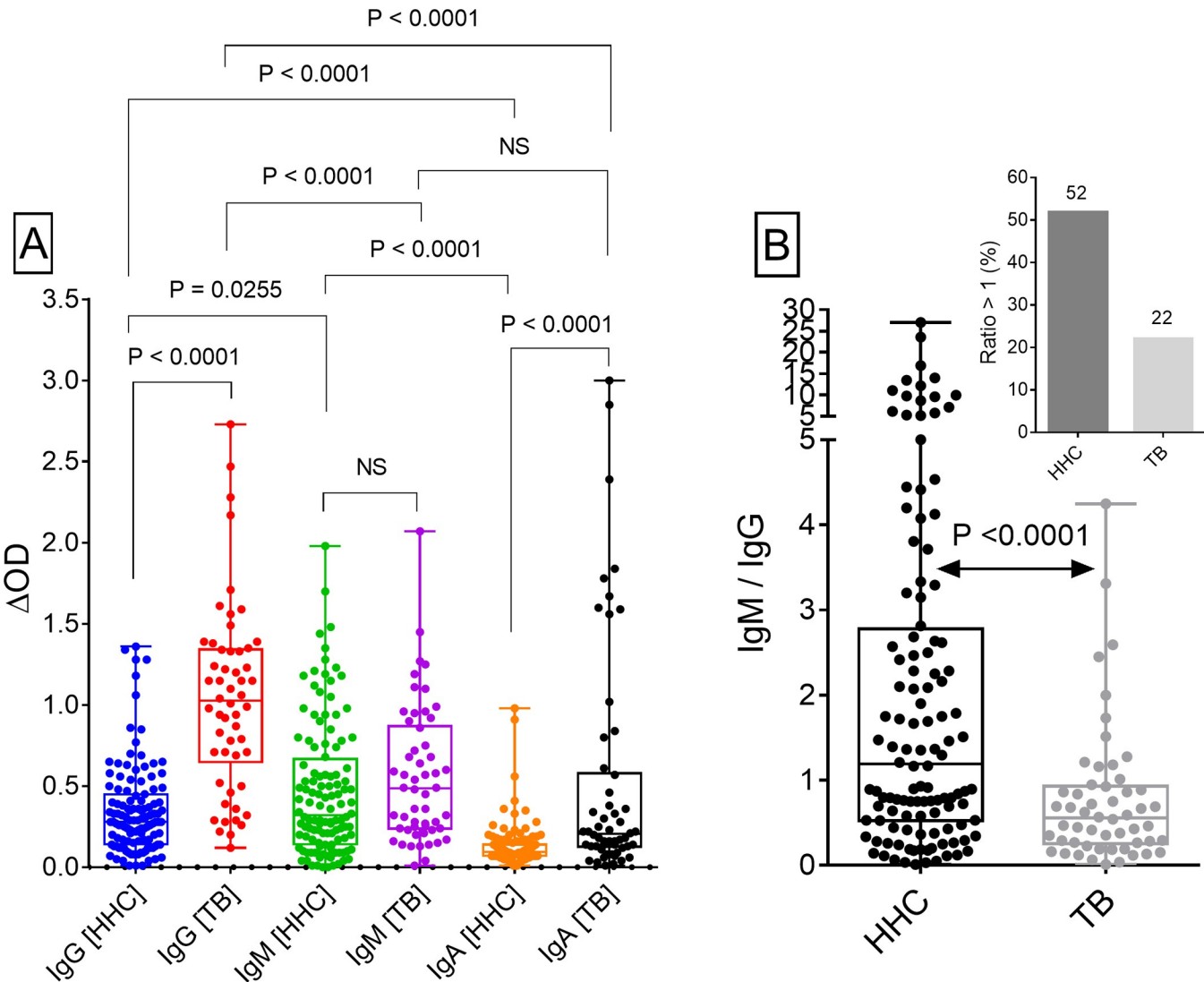

**Fig 1.** [A] Serum IgG, IgM and IgA antibody responses (ΔOD values) of HHCs and TB patients against MtM antigens. [B] IgM/IgG antibody ratios in HHCs and TB patients. Inset figure depicts proportion (%) of subjects showing a ratio of >1 (numbers on top of the bars denote % values). Statistical analyses were done by either the Wilcoxon test (for paired datasets) or Mann-Whitney test (for unpaired datasets). Two-tailed P values are shown on top of the relevant columns. NS, not significant (P > 0.05).

the levels of IgM (0.49, 0.24–0.87) or IgA (0.21, 0.13–0.58). However, among the HHCs (n = 120), levels of IgM (0.32, 0.14–0.67) were significantly higher than IgG (0.28, 0.14–0.45) or IgA (0.1, 0.07–0.14). Some more interesting observations were made on the relationship between IgM and the other two Ig isotypes. Although IgM antibody levels in HHCs and TB patients were comparable, the IgM/IgG ratio in HHCs (1.19, 0.53–2.78) was significantly higher than the ratio in patients (0.55, 0.26–0.93; Fig 1B). Consequently, this ratio was >1 in 52% of HHCs and 22% of TB patients (Fig 1B inset).

There was also a lack of correlation between IgM and IgG antibody levels in HHCs or in patients whereas the other antibody pairs (IgA vs IgM or IgA vs IgG) showed significant correlations (Fig 2). Another interesting observation was comparability of the IgM and IgA antibody levels in patients although IgM levels were significantly higher than IgA in HHCs (Fig 1). Taken together, these results suggested that IgM responses to MtM antigens dominate during the

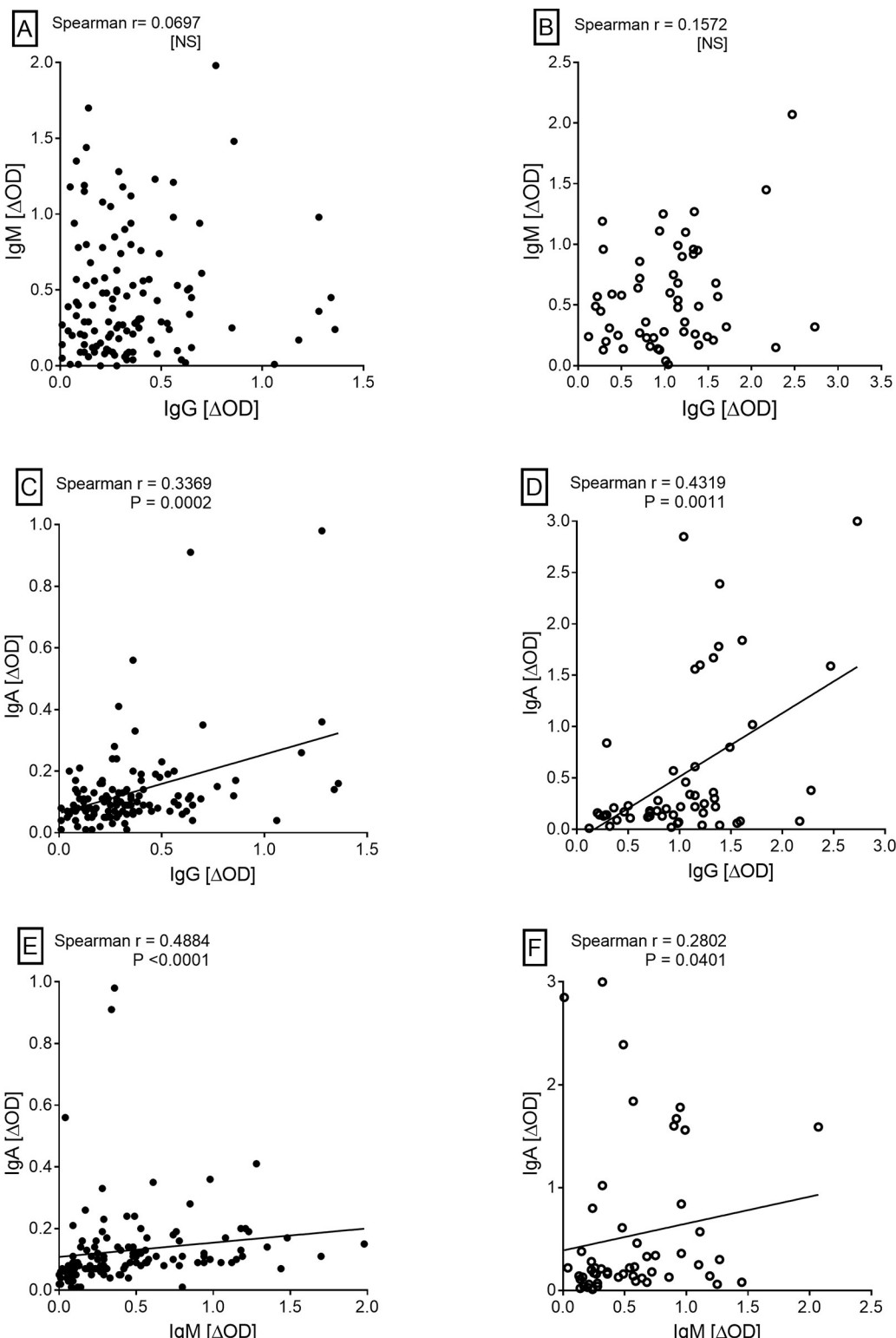

**Fig 2.** Correlation between IgG, IgM and IgA antibodies (ΔOD values) in HHCs (panels A, C and E) and TB patients (panels B, D and F). Spearman's rho and P values are shown for each XY pair. NS, not significant (P > 0.05).

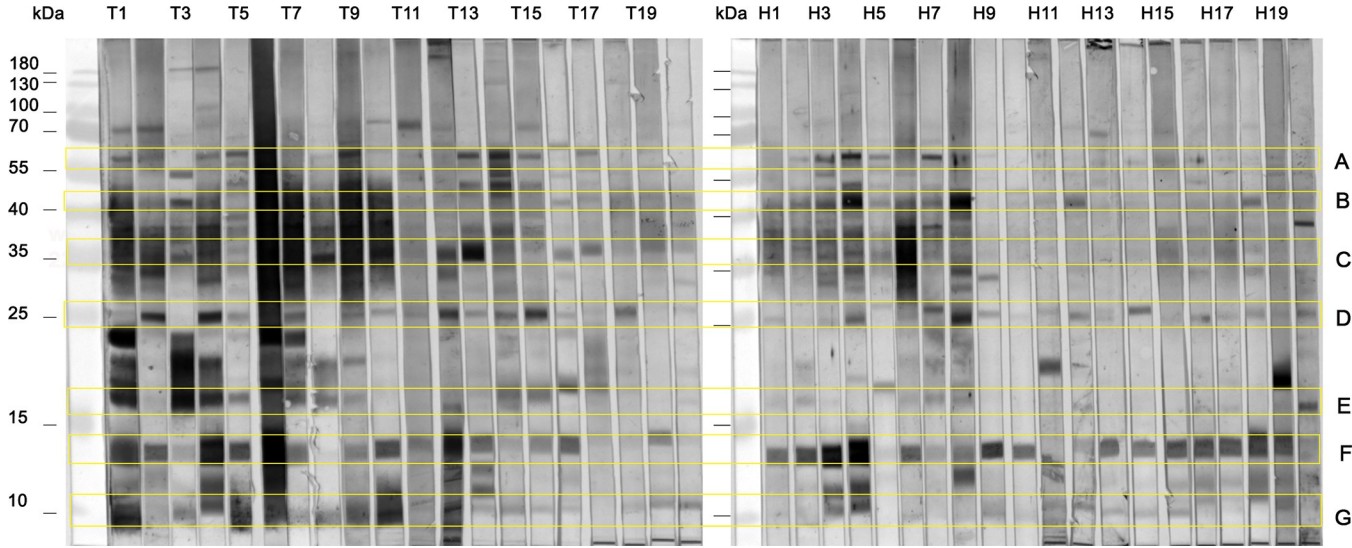

**Fig 3. Comparative immunoblot analysis of TB (n = 20, T1-T20) and HHC (n = 20, H1-H20) sera against MtM antigens.** Boxes A-G highlight antigen-antibody reactions which were shared, to a variable extent, by both the subject categories. Bands in the molecular weight range of ~25 kDa (box D), ~12 kDa (box F) and <10 kDa (box G) were shared by nearly all sera, and the other major bands (boxes A-C and E) were shared by a subset of the sera.

subclinical or latent phase of infection, whereas IgG responses dominate during the active TB disease. In addition, IgA responses could also be elevated in a subset of patients with active disease.

## Patterns of MtM antigen recognition by TB and HHC sera

Representative immunoblots of TB and HHC sera (20 each) against MtM antigens are shown in Fig 3. None of the antigens were recognised exclusively by the TB or HHC sera and there was a marked variation in the selection of antigens recognised by individual sera. Nevertheless, certain recognition patterns were discernible. Nearly all sera (TB and HHC) reacted, with a variable intensity, with antigens in the molecular wt. range of ~25 kDa (box D), ~12 kDa (box F) and <10 kDa (box G). On the other hand, only a subset of the sera reacted with antigens in the range of ~60 kDa (box A), ~45 kDa (box B) ~36 kDa (box C), and ~16 kDa (Box E). Interestingly, antigens in boxes A, B, C and E, besides being recognised by most of the patients' sera, were also recognised by some of the HHC sera (for instance, sera H1 to H8). It is unlikely that the lack of reactivity of certain HHCs sera for these antigens was due to their overall low reactivity, since they had shown a good reactivity towards other antigens (for instance, in boxes D and F). Altogether, these results suggested that certain MtM antigens could be recognised by the sera from TB patients as well as HHCs, hence corresponding antibodies could serve as the biomarkers for Mtb infection (active or latent). In addition, a subset of HHC sera could show an antigen recognition pattern similar to the TB sera.

## Frequencies of B cell subsets in peripheral blood

Frequencies of classical MBCs (cMBCs, CD19+CD27+), class-switched cMBCs (CD19+CD27+IgA+/IgG+), atypical MBCs (aMBCs, CD19+CD27-IgA+/IgG+) and plasmablasts (PBs, CD19+CD27++CD38++) in the peripheral blood of HHCs (n = 113) and TB patients (n = 47) were determined by flow cytometry using the gating strategy shown in Fig 4.

The frequency of cMBCs among lymphocytes in HHCs (median, IQR: 1.8, 1.2–2.6) was significantly higher than in patients (0.95, 0.52–1.7) (Fig 5A). Conversely, the frequency of IgG + cMBCs among cMBCs in HHCs (1.6, 0.92–2.8) was significantly lower than in patients (3,

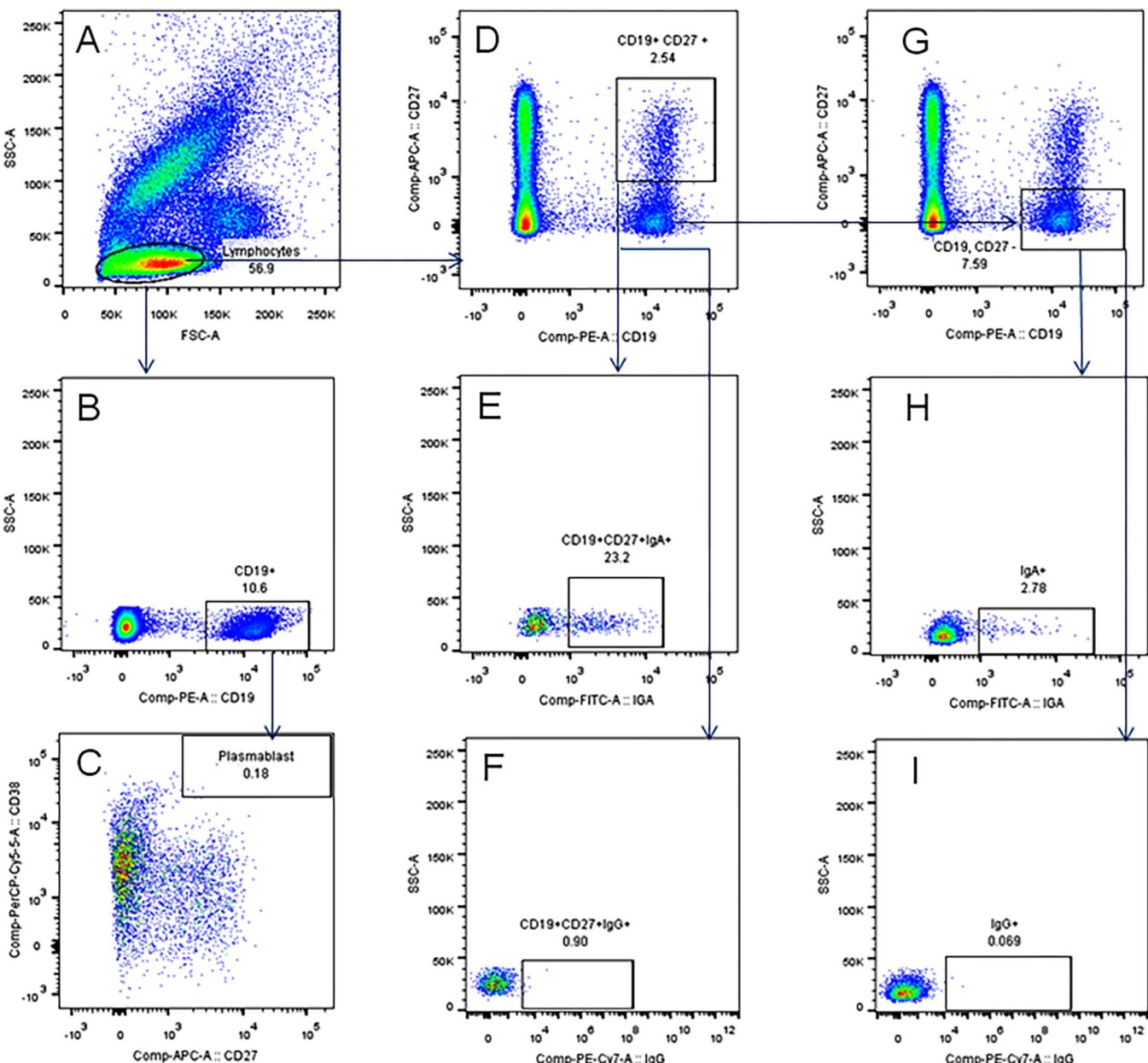

**Fig 4. Identification of B cell subsets in whole blood.** [A] RBCs were lysed, and forward (FSC) and side (SSC) scatter of WBCs was used for gating the lymphocytes. [B, C] From lymphocyte gate, B cells (stained for CD19) were gated and further analysed for staining of CD27 and CD38 in order to identify the plasmablasts (CD19+CD27++CD38++). [D] Lymphocytes were segregated on the basis of staining for CD19 and CD27 to select the classical memory B cells (CD19+CD27+) which were further segregated as IgA+ [E] and IgG+ [F] 'class-switched' classical memory B cells. The CD19+CD27- B cells [G] were further segregated as IgA+ [H] and IgG+ [I] atypical memory B cells.

1.9–5.1). Frequencies of IgA+ cMBC in HHCs (22, 12–28) and TB patients (24, 9–38) were comparable and so were the frequencies of plasmablasts among B cells (0.14, 0.08–0.26 in HHCs; and 0.23, 0.06–0.45 in TB). However, strikingly, the frequencies of IgA+ cMBCs in HHCs and TB patients were significantly higher than the corresponding frequencies of IgG+ cMBCs.

The frequencies of aMBCs (IgA+ or IgG+) among CD19+CD27- B cells in HHCs or TB patients were not significantly different. However, as observed in the case of cMBC,

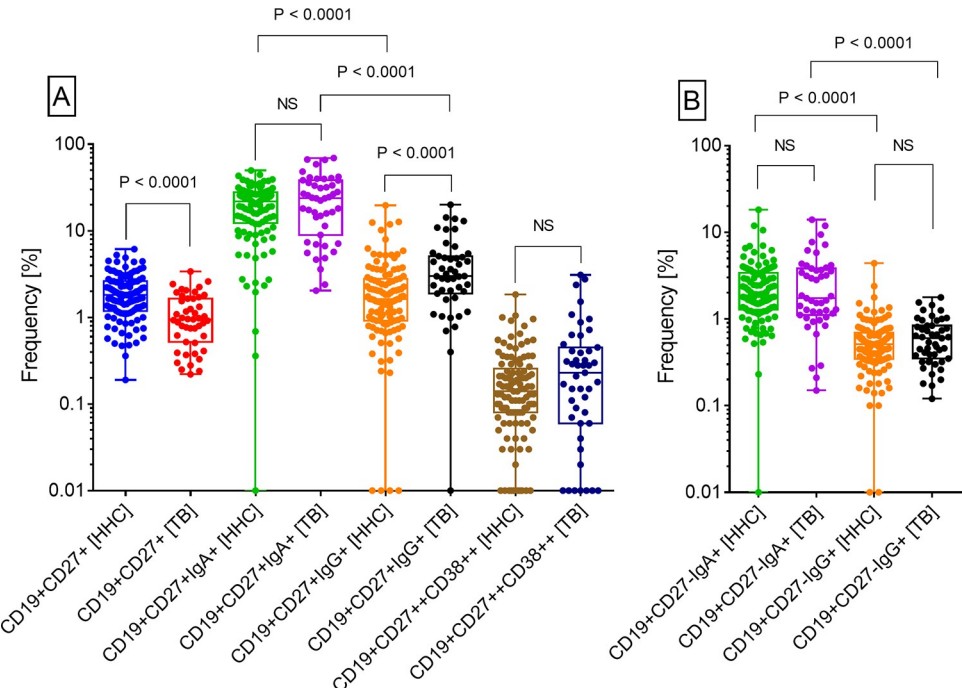

**Fig 5. Frequencies of B cell subsets in peripheral blood of TB patients (TB, n = 47) and their household contacts (HHC, n = 113).** Panel A shows frequencies of classical memory B cells (cMBCs, CD19+CD27+) as % of lymphocytes, class-switched (IgA+ or IgG+) cMBC as % of cMBCs and plasmablasts (CD19+CD27++CD38++) as % of B cells. Panel B shows frequencies of IgA+ or IgG+ atypical memory B cells as % of CD19+CD27- B cells. The gating strategy is described in Fig 4. Statistical analyses were done by either the Wilcoxon test (for paired datasets) or Mann-Whitney test (for unpaired datasets). P values for statistically significant differences are shown on top of the respective columns. NS, not significant (P > 0.05).

frequencies of IgA+ aMBCs in HHCs and patients were significantly higher than the corresponding frequencies of IgG+ aMBCs (Fig 5B). A significant correlation was seen between IgA+ cMBCs and IgA+ aMBCs in both categories of study subjects. However, such a correlation was not seen in either category for the IgG+ MBCs (Fig 6).

These results were apparently related to the quantum of the underlying infection. Relatively lower frequency of cMBCs in TB patients (as compared with HHCs) could be a consequence of their higher frequency of class-switched cMBCs. Conversely, in the absence of an active infection, HHCs exhibited a higher frequency of cMBCs and lower frequencies of other B cell types.

## Relationship between tuberculin and B cell responses

A positive TST (skin induration ≥ 10 mm) was seen in 45% of HHCs (Table 1). As evident from S1 Fig, there was no association between TST responses and the levels of antibody isotypes or frequencies of various B cell subsets. This observation reflected the discordance between cell-mediated and humoral immune responses.

## Discussion

In recent years, the quest for immunological biomarkers which could predict the outcome of a natural infection with Mtb or a vaccine against TB has turned the spotlight on B cells [4, 5]. This study was conducted on TB patients and their household contacts from India which is

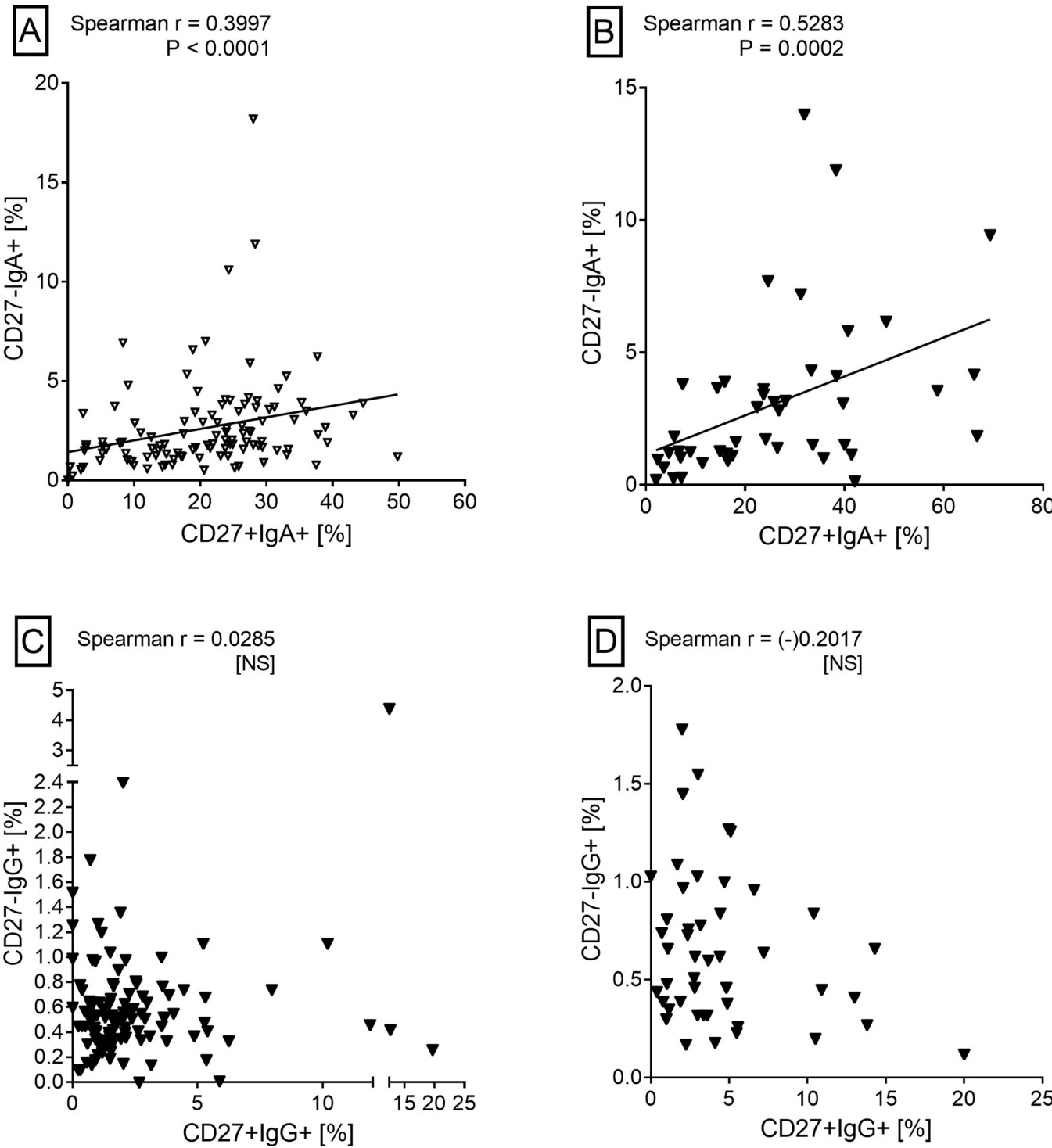

**Fig 6. Correlations between B cells subsets.** Panels A and B show significant positive correlations between IgA+ classical and atypical memory B cells in HHCs and TB patients. Panels C and D show lack of correlation between IgG+ classical and atypical memory B cells in HHCs and TB patients. Spearman's rho and P values are shown in each panel. NS, not significant (P > 0.05).

home to over a quarter of all TB patients in the world [1]. We reasoned that HHCs of smear-positive patients would be the most appropriate individuals to provide clues to the immune mechanisms involved in protection or pathogenesis following a subclinical infection with Mtb. In a landmark study from India [29], annual incidence (per 100,000 subjects) of culture-positive TB was 526 for HHCs of smear-positive patients, 271 for HHCs of smear-negative patients and 198 for non-contacts. The rate of TST positivity in our HHCs was comparable to that reported in earlier studies from India [36, 37]. For reasons not fully understood, a significant proportion of people do not respond to TST or IGRA (interferon gamma release assay) even after a prolonged and sustained exposure to patients with active TB [38]. On the other hand, high levels (up to 96% positivity) of Mtb-specific antibodies have been seen in the sera of the apparently healthy residents of TB endemic areas [38, 39]. In this study also, we noted a lack of association between B and T cell responses (the latter represented by TST) suggesting that the prevalence of LTBI in a community could be much higher than what gets estimated [3, 37].

Highest levels of anti-MtM antibodies were of IgG class in TB patients and IgM class in HHCs. Another remarkable observation was high levels of IgA antibodies in a subset of patients, leading to comparability in the IgA and IgM levels. High IgG antibody levels in TB patients have been reported in several studies and are believed to reflect the burden of infection [6, 14, 15]. High IgA levels against certain Mtb antigens have also been reported in patients [40]. The dominant IgM responses in LTBI have been reported previously by us as well as others. We had shown high anti-MtM IgM levels in healthy Indian subjects exposed to Mtb [16]. In another study, subjects with LTBI harbored high IgM levels against the antigens of Mtb lysate (including MtM) [41]. In yet another study, contacts of cavitary TB patients had high IgM levels against Mtb culture filtrate antigens [42]. These studies suggest that the dominance of IgM response in HHCs is not dependent on the nature of Mtb antigen. It would be pertinent to consider these results in the context of natural course of Mtb infection which is essentially airborne. Initial exposure to Mtb (e.g., in HHCs) is likely to trigger the production of IgM, which is the first antibody released during a humoral immune response. Following its release by the plasma cells of lymphoid organs, IgM enters mucosal secretions as secretory IgM (sIgM) [43] which may provide protection at an early stage of infection until, and as a result of increasing burden of infection, a progressively dominant production of sIgA takes over [44]. IgG is more abundant than IgA in lower respiratory tracts and its concentration in the upper respiratory tracts increases with the rising infection. In all likelihood, IgG also exerts its defensive functions in the airways in concert with sIgA and sIgM [43]. The mucosal and systemic immune compartments are not mutually exclusive and cells, antibodies or antigens can travel from one to the other compartment through blood and lymph. Antigen-specific antibodies produced through canonical T-dependent B cell responses can facilitate the elimination of bacteria by opsonisation followed by phagocytosis [8, 9]. The results of this and earlier studies suggest that, with increasing burden of infection, antibody responses may shift from a predominantly IgM to predominantly IgG type. This impression also gains support from the lack of correlation between IgM and IgG antibody levels and a significantly higher IgM/IgG ratio in HHCs than in patients. Similar findings have been reported in another study [41] suggesting that a progressive decline in IgM/IgG antibody ratio in HHCs could potentially indicate a rise in the infection level.

MtM antigens of interest, as determined by immunblotting, fell into two categories: those (e.g., of <10, ~12 and ~25 kDa) recognised, although with a variable intensity, by all tested sera (TB and HHC) and those (e.g., of ~16, ~36, ~45 and ~60 kDa) recognised by almost all TB sera and a subset of HHC sera. In previous studies also these antigens have been identified as major seroreactive antigens of Mtb, although results of each study were different from the other [13, 17–21]. There could be several reasons for the discordance between present and past

studies. Most probable reason could be the differences in demography of study subjects as we have studied HHCs of only the smear-positive patients. Another reason could be the nature of Mtb antigens used for immunoblotting. Most studies have used whole cell lysate or culture filtrate antigens, although a study from India [19] has also looked at MtM antigens. Additionally, in some studies [13, 17, 21] the proteins were salted-out prior to use. There are also variations is the molecular weight window within which the antigens were visualised, viz., 3–45 kDa [21], 20–97 kDa [20] or ≥12 kDa [19]. Our immunoblot results have two important implications: First, it may be possible to detect the Mtb infection (active or latent) based on seroreactivity for selected MtM antigens. Second, the HHCs showing an antigen-recognition pattern similar to TB patients could be harbouring an elevated burden of infection. The progression from latent to active TB disease is believed to be accompanied by qualitative as well as quantitative changes in the bacterial antigen expression [15].

The frequency of classical MBCs (CD27+ B cells) was significantly higher in HHCs than in patients. In some earlier studies also [9, 25], subjects with LTBI had shown significantly higher cMBCs than the patients. cMBCs are precursors of the class-switched cMBCs which are regarded as true MBCs since, upon re-challenge with antigen, they can rapidly transform into antibody secreting cells. IgG+ cMBCs are considered as the most prominent subset of switched cMBCs [45]. Accordingly, we also found that the frequency of IgG+ cMBC in patients was significantly higher than in HHCs. There is no prior study on the frequency of IgG+ MBCs in TB or HHC. However, Ashenafi et al [26] have reported a significantly higher frequency of IgG + PBs in active TB than in LTBI. Remarkably, the frequencies of IgA+ cMBCs in HHCs and patients were significantly higher than IgG+ cMBCs. Soe et al [25] have also shown that the IgA+ cMBCs were significantly higher than IgG+ cMBCs in subjects with LTBI, although they did not perform this analysis for TB patients. Mtb infection is airborne in nature and airway responses are typically IgA biased [44]. Hence, a significant IgA+ cMBC expansion early during infection may play a crucial role in preventing its spread [9]. It is also possible that IgA + MBCs generated during the mucosal immune response migrate from lungs to the periphery [9]. In the steady state, a majority of IgA+ cMBCs in peripheral blood co-express the mucosal homing receptors CCR10 and α4β7 indicating that they had originated from the mucosal immune response [46]. As seen for IgA+ cMBCs, frequencies of IgA+ atypical MBCs in HHCs and TB patients were also significantly higher than the corresponding IgG+ aMBCs. Moreover, a significant correlation was seen between IgA+ cMBCs and IgA+ aMBCs in both the study groups. IgA+ aMBCs have been shown to arise from T-independent B cell response and antibodies produced by them bind promiscuously to bacterial carbohydrates [47]. Mtb cell envelope has an abundance of such complex carbohydrates [48] hence IgA+ aMBCs could, along with IgA+ cMBCs, be a part of the mucosal defence mechanism against Mtb [22]. Although no significant information is available on the role of IgG+ aMBCs, they could also be a part of the mucosal immune-surveillance [22, 49]. We did not find a significant difference in the frequencies of PBs in HHCs and TB patients although an earlier study had shown significantly higher frequency of PBs in TB than in LTBI [9]. The most probable reason for this discordance could be the use of a much larger cohort and relatively higher exposure of the HHCs to Mtb infection in our study.

Our study has some important limitations. Firstly, due to its cross-sectional nature we could not validate the B cell responses suggestive of persistence or reactivation of LTBI. To draw some definitive conclusions in this regard, a longitudinal study of sufficiently large cohort over a follow-up period of several years is required [29]. Secondly, we have not determined the frequency of 'IgM only' (IgM+IgD-) MBCs since our aim was to profile the major class-switched (IgG+ and IgA+) MBCs which are generally regarded as true B-cell memory [45]. 'IgM only' memory is also considered as a 'pre-switched' state capable of undergoing

further class switch [22]. Two issues have hampered the studies on IgM only MBCs [23]: the IgM+IgD- and IgM+IgD+ subsets have not always been separated, and the IgD- population is often not subdivided. Hence, one gets a mixed population of class-switched and IgM+ memory B cells. Du Plessis et al [50] had shown that the frequencies of IgM+ cMBC and IgM+ PBs were significantly higher in untreated than in treated TB patients and controls. However, the authors had focused only on IgM+ and IgM− B cells. The immunoglobulin isotypes of IgM − cells were not determined. They also did not determine whether or not the IgM+ cells co-expressed IgD.

To conclude, this study highlights the differences and similarities between B cell responses of smear-positive TB patients and their household contacts. The results suggest that some of the responses, for instance, the ratio of IgM to IgG antibodies, antibody binding to selected antigens, and relative frequencies of memory and switched memory B cells could serve as indicators of protective or pathogenic immune responses following the initial infection with Mtb. This study also supports the view that elicitation of mucosal immunity could be an important goal for a vaccine against TB [51].

## Supporting information

**S1 Raw images. Original unprocessed images for Fig 3.**
(PDF)

**S1 Fig. Relationship between tuberculin and B cell responses.**
(PDF)

**S1 Table. Raw data for Fig 1, 2, 5 and 6.**
(PDF)

## Acknowledgments

We are grateful to the patients and their families for agreeing to participate in this study.

## Author Contributions

**Conceptualization:** Sudhir Sinha, Amita Aggarwal.

**Data curation:** Sudhir Sinha.

**Formal analysis:** Sudhir Sinha.

**Funding acquisition:** Amita Aggarwal.

**Investigation:** Komal Singh, Rajesh Kumar, Fareha Umam, Prerna Kapoor.

**Methodology:** Sudhir Sinha.

**Project administration:** Sudhir Sinha, Amita Aggarwal.

**Resources:** Sudhir Sinha, Amita Aggarwal.

**Supervision:** Sudhir Sinha, Amita Aggarwal.

**Validation:** Amita Aggarwal.

**Visualization:** Sudhir Sinha.

**Writing – original draft:** Sudhir Sinha.

**Writing – review & editing:** Amita Aggarwal.

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
