## [Decision Letter · Decision Letter 0]

4 Oct 2022

PONE-D-22-24848Distinct and shared B cell responses of tuberculosis patients and their household contactsPLOS ONE

Dear Dr. Aggarwal,

Thank you for submitting your manuscript to PLOS ONE. After careful consideration, we feel that it has merit but does not fully meet PLOS ONE’s publication criteria as it currently stands. Therefore, we invite you to submit a revised version of the manuscript that addresses the points raised during the review process.

We look forward to receiving your revised manuscript.

Kind regards,

Juraj Ivanyi

Academic Editor

PLOS ONE

Reviewers' comments:

Reviewer's Responses to Questions

**Comments to the Author**

1. Is the manuscript technically sound, and do the data support the conclusions?

Reviewer #1: Yes

Reviewer #2: Yes

2. Has the statistical analysis been performed appropriately and rigorously? 

Reviewer #1: Yes

Reviewer #2: Yes

3. Have the authors made all data underlying the findings in their manuscript fully available?

Reviewer #1: Yes

Reviewer #2: Yes

4. Is the manuscript presented in an intelligible fashion and written in standard English?

Reviewer #1: Yes

Reviewer #2: Yes

5. Review Comments to the Author

Reviewer #1: This original research article characterised the B cell responses in tuberculosis (TB) patients and their household contacts, with the view of determining whether these responses can distinguish between latent and active TB disease. The authors recruited smear-positive TB patients and their disease-free household contacts. Serum samples were acquired to determine antibody levels against various Mycobacterium tuberculosis (Mtb) membrane antigens. Whole-blood samples were also taken to characterise populations of B cell subsets by flow cytometry. Briefly, the study found that TB patients had higher levels of IgG against Mtb membrane antigens, whereas household contacts had higher levels of IgM. Interestingly, the authors found differences in the reactivity of TB patient and household contact sera against Mtb membrane antigens which was visualised by immunoblotting. Furthermore, the authors found differences in the frequencies of classical memory B cells (CD19+CD27+) and between TB patients and their household contacts, while frequencies of IgA+ atypical memory B cells (CD19+CD27-) were higher than IgG+ in both TB patients and household contacts. The article concludes that the differences in B cell responses between TB patients and household contacts observed could play a role in a protective immune response against TB, while providing useful information for correlates of protection in vaccine development.

Overall, the article was well-written with a good quality of English and ample detail was provided regarding the methodology. The absence of a “true” negative control group was concerning. While the conclusions made in this article are still valid despite the lack of this control group, a stronger case could be made if this data could be included.

Major comments

• The data and figures in this study would benefit greatly from inclusion of negative controls (samples from TB negative individuals who are not household contacts of active TB patients, whether BCG vaccinated or not). This would allow for further discussion of confounding factors such as exposure to environmental mycobacterial antigens. It would also add weight to specificity argument when discussing immune reactivity to different protein fractions of membrane proteins.

• The authors should discuss the implication of BCG status (I presume most subjects had a BCG scar) on their interpretations of data

• I appreciate the authors’ comments about their data being useful in the context of diseases progression and vaccine induced protective immunity. Is it not, however, that some of the higher molecular weight proteins might be the candidates for serologically distinguishing TB cases and latent TB (diagnostic value)?

Minor comments

• Line 60. Sentence should read “The role of antibodies in protection…”

• Line 176. Include make/model of plate reader used

• Line 212. Include version number for GraphPad Prism

• Table 1. Check spelling, 2 should read “GeneXpert”

Reviewer #2: In this paper, the authors evaluated, in a cohort of tuberculosis patients and their smear-positive household contacts, the humoral adaptive immune compartment by measuring Mtb-membrane-specific antibody levels. Moreover, they also evaluated the B cell subsets in peripheral blood.

Results showed in TB patients a significantly higher level of IgG than IgM and IgA whereas IgM and IgA antibody levels were comparable. In HHC sera had significantly higher IgM antibody levels than IgG and IgA. The ratio of IgM to IgG antibodies in HHCs was also significantly higher than in TB patients. Immunoblotting displays that HHC and TB patients shared the ability to recognize specific low-molecular weight MtM antigens. About B cells, the classic memory B Cells Frequencies were significantly higher, and IgG+ cMBCs were significantly lower in HHCs than in patients.

The authors conclude that IgM/IgG antibody ratio, antibody binding to selected MtM antigens and relative frequencies of MBC subsets could indicate protective or pathogenic immune responses following the primary infection with Mtb.

In my opinion, the paper is well written and gives interesting ideas about the still controversial role of humoral response in Mtb infection and tuberculosis.

I have some points to observe about the results and discussion.

In the legend of figure 1 and 5, the authors should add the statistic test used.

About the statistic, considering the number of patients and HHC enrolled, the authors can show data using the mean and not the median and can use parametric statistic tests if they have a normal data distribution.

The period between lines 298 and 304 is not fully clear. For instance, the authors write that the median frequency of cMBC is 1.8 in HHC and 0.95 in TB patients, but after they write that the frequency of IgG+ cMBC is 1.6 in HHC and 3.0 in TB patients, the last value is higher than 0.95. In my opinion, even if they show a gating strategy, the authors should better specify from which populations the reported frequencies are counted or should report the frequencies to the whole B cells population.

Line 349, are the incidence results based on one hundred thousand subjects per year? Please specify.

Line 353 have the authors checked if there is any correlation between the TST responders among HHCs and the concentration of IgM or IgG or IgA or any other parameter related to B cells?

Line 358, the authors used the LTBI abbreviation for the first time in the paper; they should give the full words of this abbreviation.

Line 365, in literature, there are many discordant data about the concentration and role of antibodies and B cells in tuberculosis or TB infection. Therefore, the authors should give a more thorough background on this point.

Line 407, the authors write: “the HHCs showing an antigen-recognition pattern similar to TB patients could be harbouring an elevated burden of infection.” This is an interesting point. Do the authors plan to verify this hypothesis?

6. PLOS authors have the option to publish the peer review history of their article (what does this mean?). If published, this will include your full peer review and any attached files.

Reviewer #1: No

Reviewer #2: No

---

## [Author Response · Author response to Decision Letter 0]

8 Oct 2022

Response to Reviewers

Reviewer #1: 

General comments: This original research article characterised the B cell responses in tuberculosis (TB) patients and their household contacts, with the view of determining whether these responses can distinguish between latent and active TB disease. The authors recruited smear-positive TB patients and their disease-free household contacts. Serum samples were acquired to determine antibody levels against various Mycobacterium tuberculosis (Mtb) membrane antigens. Whole-blood samples were also taken to characterise populations of B cell subsets by flow cytometry. Briefly, the study found that TB patients had higher levels of IgG against Mtb membrane antigens, whereas household contacts had higher levels of IgM. Interestingly, the authors found differences in the reactivity of TB patient and household contact sera against Mtb membrane antigens which was visualised by immunoblotting. Furthermore, the authors found differences in the frequencies of classical memory B cells (CD19+CD27+) and between TB patients and their household contacts, while frequencies of IgA+ atypical memory B cells (CD19+CD27-) were higher than IgG+ in both TB patients and household contacts. The article concludes that the differences in B cell responses between TB patients and household contacts observed could play a role in a protective immune response against TB, while providing useful information for correlates of protection in vaccine development. Overall, the article was well-written with a good quality of English and ample detail was provided regarding the methodology.

Response: We are grateful for these very encouraging comments. 

Comment: The absence of a “true” negative control group was concerning. While the conclusions made in this article are still valid despite the lack of this control group, a stronger case could be made if this data could be included.

Response: This response is clubbed with the response to comment-1 (below).

Major comments

Comment-1: The data and figures in this study would benefit greatly from inclusion of negative controls (samples from TB negative individuals who are not household contacts of active TB patients, whether BCG vaccinated or not). This would allow for further discussion of confounding factors such as exposure to environmental mycobacterial antigens. It would also add weight to specificity argument when discussing immune reactivity to different protein fractions of membrane proteins.

Response: Indeed the data could have benefitted from the inclusion of negative controls. However, the worldwide prevalence of TB suggests that it is difficult to find persons who have not been exposed to Mtb. Even in studies (for instance, the ones cited for immunoblotting) which had included ‘healthy’ or ‘non TB’ controls (defined by a negative TST or IGRA), a large proportion of the controls actually ended up in showing an immune response to Mtb antigens. As mentioned in the first paragraph of Discussion, a significant proportion of people do not respond to TST or IGRA even after a prolonged and sustained exposure to patients with active TB. In fact, up to 96% of healthy residents of TB endemic areas show Mtb-specific antibodies in their sera. This situation is particularly relevant for the ‘high TB burden’ countries like India. According to the Indian study [Ref 29] cited in the Discussion (1st paragraph), a good number of healthy subjects who developed TB during the follow-up period had no known contact with a TB patient. In another study from north India, the incidences of TB emanating from TST/IGRA positive and negative contacts were comparable suggesting that both tests could be missing a large proportion of Mtb-infected subjects (Sharma et al, PLoS ONE, 2017; 12: e0169539). The fact that we could be underestimating the true extent of Mtb infection in the community is highlighted by a recent article (Dowdy and Behr, Lancet Infect Dis, 2022 Sep: e271-e278). 

Comment-2: The authors should discuss the implication of BCG status (I presume most subjects had a BCG scar) on their interpretations of data.

Response: All study subjects were adults, residing in semi-urban or rural areas of North India and only a small proportion of them had received the BCG vaccine during infancy (according to the vaccination policy in India). Although the immunization coverage in semi-urban/rural India has expanded over the years, a coverage gap still exists among children between poor and rich households or between those belonging to educated and uneducated mothers (Khan and Saggurti, Vaccine. 2020; 38: 4088-4103). With respect to the effect of BCG vaccine on antibody responses, in a study from India (Das et al, Clin Exp Immunol.1992; 89:402-6) the BCG vaccination status was found to have no effect on the antibody levels against Mtb antigens in either Indian or British subjects. 

Comment-3: I appreciate the authors’ comments about their data being useful in the context of diseases progression and vaccine induced protective immunity. Is it not, however, that some of the higher molecular weight proteins might be the candidates for serologically distinguishing TB cases and latent TB (diagnostic value)?

Response: Thanks for this suggestion which we would like to explore in our future studies. However, we feel that our present data is insufficient for making this claim. 

Minor comments

• Line 60. Sentence should read “The role of antibodies in protection…” [Response: Corrected]

• Line 176. Include make/model of plate reader used [Response: Included]

• Line 212. Include version number for GraphPad Prism [Response: Included]

• Table 1. Check spelling, 2 should read “GeneXpert” [Response: Corrected]

Reviewer #2: 

General comments: In this paper, the authors evaluated, in a cohort of tuberculosis patients and their smear-positive household contacts, the humoral adaptive immune compartment by measuring Mtb-membrane-specific antibody levels. Moreover, they also evaluated the B cell subsets in peripheral blood. Results showed in TB patients a significantly higher level of IgG than IgM and IgA whereas IgM and IgA antibody levels were comparable. In HHC sera had significantly higher IgM antibody levels than IgG and IgA. The ratio of IgM to IgG antibodies in HHCs was also significantly higher than in TB patients. Immunoblotting displays that HHC and TB patients shared the ability to recognize specific low-molecular weight MtM antigens. About B cells, the classic memory B Cells Frequencies were significantly higher, and IgG+ cMBCs were significantly lower in HHCs than in patients. The authors conclude that IgM/IgG antibody ratio, antibody binding to selected MtM antigens and relative frequencies of MBC subsets could indicate protective or pathogenic immune responses following the primary infection with Mtb. In my opinion, the paper is well written and gives interesting ideas about the still controversial role of humoral response in Mtb infection and tuberculosis. I have some points to observe about the results and discussion.

Response: We are grateful for these very encouraging comments. 

Comment-1: In the legend of figure 1 and 5, the authors should add the statistic test used.

Response: These additions have been made. 

Comment-2: About the statistic, considering the number of patients and HHC enrolled, the authors can show data using the mean and not the median and can use parametric statistic tests if they have a normal data distribution.

Response: We used median (along with IQR) because the data distribution was skewed. For the same reason, we used the non-parametric tests.

Comment-3: The period between lines 298 and 304 is not fully clear. For instance, the authors write that the median frequency of cMBC is 1.8 in HHC and 0.95 in TB patients, but after they write that the frequency of IgG+ cMBC is 1.6 in HHC and 3.0 in TB patients, the last value is higher than 0.95. In my opinion, even if they show a gating strategy, the authors should better specify from which populations the reported frequencies are counted or should report the frequencies to the whole B cells population.

Response: Thanks for pointing out this important issue which indeed needed resolution. We have now modified the legends to Figures 4 and 5, as well as the text in the relevant Results section. Hopefully these modifications have clarified the matter. 

Comment-4: Line 349, are the incidence results based on one hundred thousand subjects per year? Please specify.

Response: Yes, indeed the incidence rates are per 100,000 subjects/per year. We have now added this information in the revised text.

Comment-5: Line 353 have the authors checked if there is any correlation between the TST responders among HHCs and the concentration of IgM or IgG or IgA or any other parameter related to B cells?

Response: Yes, we have described this data in the Results section and have also shown it as Supplementary Figure S2.

Comment-6: Line 358, the authors used the LTBI abbreviation for the first time in the paper; they should give the full words of this abbreviation.

Response: In fact, we have used it (along with full words) in the first paragraph of Introduction. 

Comment-7: Line 365, in literature, there are many discordant data about the concentration and role of antibodies and B cells in tuberculosis or TB infection. Therefore, the authors should give a more thorough background on this point.

Response: As suggested, we have now revised the Introduction (2nd paragraph). Accordingly, we have also replaced Ref no. 6 with a more relevant review article.

Comment-8: Line 407, the authors write: “the HHCs showing an antigen-recognition pattern similar to TB patients could be harbouring an elevated burden of infection.” This is an interesting point. Do the authors plan to verify this hypothesis?

Response: Thanks for pointing this out. This is indeed an interesting yet largely unexplored area. It is interesting because immunoblotting represents an unbiased, hypothesis-free approach for the detection of serologically active immunodominant antigens. We have plans to generate a sufficiently large dataset in order to draw some robust conclusions.

---

## [Editor Report · Decision Letter 1]

11 Oct 2022

Distinct and shared B cell responses of tuberculosis patients and their household contacts

PONE-D-22-24848R1

Dear Dr. Aggarwal,

We’re pleased to inform you that your manuscript has been judged scientifically suitable for publication and will be formally accepted for publication once it meets all outstanding technical requirements.

Kind regards,

Juraj Ivanyi

Academic Editor

PLOS ONE
---

## [Editor Report · Acceptance letter]

14 Oct 2022

PONE-D-22-24848R1 

Distinct and shared B cell responses of tuberculosis patients and their household contacts 

Dear Dr. Aggarwal:

I'm pleased to inform you that your manuscript has been deemed suitable for publication in PLOS ONE. Congratulations! Your manuscript is now with our production department. 

Kind regards, 

on behalf of

Dr. Juraj Ivanyi 

Academic Editor

PLOS ONE